# Epidemiological baseline of *Brucella* spp. in South African wildlife

Carlo Andrea Cossu[1]*, Jeanette Wentzel[2], Lin-Mari de Klerk[3], Fabrizio De Massis[4], Jacques Godfroid[5], Louis Ockert van Schalkwyk[3], Giuliano Garofolo[4], Henriette van Heerden[1]

1 Department of Veterinary Tropical Diseases, Faculty of Veterinary Science, University of Pretoria, Onderstepoort, South Africa, 2 Wildlife Studies, Faculty of Veterinary Science, University of Pretoria, Onderstepoort, South Africa, 3 Office of the State Veterinarian, Department of Agriculture, Land Reform and Rural Development, Skukuza, South Africa, 4 National and WOAH reference laboratory for brucellosis, Istituto Zooprofilattico Sperimentale dell'Abruzzo e del Molise "G. Caporale", Teramo, Italy, 5 Department of Arctic and Marine Biology, Faculty of Biosciences, Fisheries and Economics, UiT – The Arctic University of Norway, Tromsø, Norway

* ca.cossu@tuks.co.za

## Abstract

Brucellosis is a globally significant zoonotic disease, yet its ecology in wildlife remains poorly understood. In this study, we investigated the circulation of *Brucella* spp. in a wide range of wild mammals from multiple protected areas in South Africa. Organ and serum samples from 722 animals representing 23 species were analyzed employing a conservative diagnostic strategy, combining serology (rose bengal test confirmed by indirect ELISA) and four PCR-based assays in series, to maximize specificity and interpretative solidity. Molecular detection revealed *Brucella* spp. DNA in several atypical host species, including plains zebra, hippopotamus, African elephant, giraffe, warthog, cheetah, and African wild dog, expanding the known host range. In Greater Kruger National Park, African buffalo exhibited high seroprevalence (24/106; 23%, CI: 15–32%) and even higher molecular prevalence (29/57; 51%, CI: 37–64%), supporting their role as a primary wildlife reservoir for *B. abortus* in the region. One *B. abortus* isolate cultured from the spleen of a zebra and characterized by multiloci variable number of tandem repeat analysis (MLVA) showed genetic relatedness to South African buffalo and South American cattle strains. Co-infections with *B. abortus* and *B. melitensis* were identified in 17 animals across seven species. Notably, *B. melitensis* was detected in species (elephant, hippopotamus, zebra) not typically associated with small ruminants, suggesting complex interspecies transmission dynamics. Our findings underscore the limitations of serological testing and highligts the value of molecular diagnostics in understanding the epidemiology of *Brucella* spp. in South Africa. The detection of *B. abortu*s and/or *B. melitensis* DNA in a range of wildlife species, including carnivores and megaherbivores, emphasises the

**Data availability statement:** All the sequences obtained in this study were deposited in GenBank database under the accession numbers: PX108534-PX108543. All other relevant data are in the manuscript and its supporting information files.

**Funding:** This research was supported by funding from agriSETA (to HvH) and the Belgian Directorate-General for Development Cooperation through its Framework Agreement with the Institute for Tropical Medicine [FA4 DGD-ITM 2017-2021 & FA5 DGD-ITM 2022-2026 to HvH]. The funders had no role in study design, data collection and analysis, decision to publish, or preparation of the manuscript. The authors received no specific funding for this work.

**Competing interests:** The authors have declared that no competing interests exist.

need for integrated One Health surveillance approaches to enhance understanding of the disease's life cycle and transmission dynamics.

## Author summary

Brucellosis is an infectious disease that affects both animals and humans, causing reproductive problems in livestock and long-term illness in people. While much is known about the disease in cattle, sheep and goats, little is understood about its circulation in wild animals. In this study, we explored how *Brucella* bacteria spread among different wildlife species in South Africa. We tested blood and tissue samples from a wide variety of wild animals, including buffalo, zebra, elephants and various bovids and carnivores, using both blood antibody tests and DNA-based methods to ensure reliable results. We found signs of infection not only in buffalo, already known to carry *Brucella abortus*, but also in unexpected hosts such as zebra, giraffe, hippopotamus and elephants. We also detected *Brucella melitensis*, a species typically associated with goats and sheep, in several wild species. These findings show that *Brucella* bacteria circulate more widely in African wildlife than previously thought, and that infections may move between domestic and wild animals. Our study highlights the importance of using modern molecular tools and a One Health approach, linking animal, human, and environmental health, to better understand and manage diseases shared between people and wildlife.

## Introduction

Brucellosis is an important animal disease and one of the most prevalent zoonoses worldwide, especially in resource-limited settings [1]. It is caused by intracellular Alphaproteobacteria of the genus *Brucella*, with twelve species identified up-to-date: *B. abortus* from cattle, *B. melitensis* from sheep and goats [2–4], *B. ovis* from sheep [5], *B. suis* from domestic and wild suids [6], *B. canis* from dogs [7], *B. neotomae* from woodrats (*Neotoma* spp.) [8], *B. inopinata* from human samples [9], *B. microti* from common voles (*Microtus* spp.) [10,11], *B. pinnipedialis* and *B. ceti* from marine mammals [12], *B. papionis* from baboons (*Papio* spp.) [13] and *B. vulpis* from red foxes (*Vulpes vulpes*) [14]. Recent phylogenetic analyses have revealed that the separation between former *Ochrobactrum* spp. are not distinct from *Brucella* spp. as previously thought [15]. This close relationship, supported by whole genome or 16S rRNA analysis, has led to the recent taxonomic reclassification of the two genera into a single expanded *Brucella* genus [16]. However, this has generated a lot of confusion to the point that it has been urged to keep Brucella and *Ochrobactrum* genera separate to avoid further bewilderment and harm [17].

Brucella abortus and *B. melitensis* are the two main species occurring in South Africa [18], raising concerns for their economic damages to the agricultural sector

and zoonotic potential. *Brucella abortus* is a major pathogen in cattle, causing abortions, stillbirths, infertility, and mastitis, which lead to reduced milk production and significant economic losses [19]. Worldwide, most human cases are caused by *Brucella melitensis*. As highly pathogenic bacteria, *Brucella* spp. suspect samples and isolates must be handled in biosafety level 3 laboratories to ensure the safety of laboratory personnel.

Globally, the disease remains a major public health concern, with an estimated 1.6–2.1 million new human cases annually, primarily in Asia (1.2–1.6 million cases) and Africa (0.5 million cases) [1]. Human brucellosis is challenging due to its potential to cause chronic, debilitating illness. The bacteria's ability to persist in a dormant state and relapse years after initial infection [20] further complicates diagnosis and treatment, which may require prolonged or even lifelong management.

Most human cases occur in areas with high population density and limited access to veterinary and public health infrastructure [1]. Ongoing global population growth, especially in Africa, amplifies these risks [21]. Moreover, in many African countries, weak or nonexistent disease surveillance systems and inadequate animal health programs contribute to the persistence of brucellosis [22]. Environmental change and expanding human encroachment into wildlife habitats increasingly facilitate the spillover of *Brucella* between and within domestic animals and wildlife [23–26]. This dynamic not only undermines livestock production and food security but also poses a growing threat to conservation efforts and public health, particularly in rural communities that rely on livestock.

In South Africa, it has been suggested that *Brucella* originally spilled over from domestic livestock to wild buffalo populations through cattle trade, eventually establishing an endemic persistence in ecosystems such as Kruger National Park [24]. Transmission among wild ungulates likely occurs through ingestion of contaminated forage, while carnivores may acquire infection by consuming infected tissues, including aborted fetuses, placentas, or carcasses [17].

Brucellosis surveillance in wildlife primarily relies on serological testing that has been validated for livestock species [26]. Moreover, serological tests do not confirm active infection nor identify the specific *Brucella* species. Serological evidence of exposure to *Brucella* spp. has been reported in a wide range of African wildlife. This includes wild bovids such as African buffalo (*Syncerus caffer*) [27–34], blue wildebeest (*Connachaetes taurinus*) [28,35,36], eland (*Taurotragus oryx*) [29,31], impala (*Aepyceros melampus*) [31,32,37], kafue lechwe (*Kobus leche kafuensis*) [38], kudu (*Tragelaphus strepsiceros*) [23], and waterbuck (*Kobus ellipsiprymnus*) [37]. Evidence has also been found in carnivores such as black-backed jackal (*Lupulella mesomelas*) [35], lion (*Panthera leo*) [29], leopard (*Panthera pardus*) [29], spotted hyena (*Crocuta Crocuta*) [35], as well as atypical hosts like giraffe (*Giraffa camelopardalis*) [27,29,31,39], hippopotamus (*Hippopotamus amphibius*) [37], warthog (*Phacochoerus aethiopicus*) [29] and plains zebra (*Equus quagga*) [35]. While these tests can detect exposure in wildlife, it remains uncertain whether they are effective across all host species.

On the other hand, only a few studies focused on the molecular detection of *Brucella* spp. in wildlife. Sambu et al. [40] detected *Brucella* DNA in 7/46 (15%; CI: 6–29%) buffalo, 3/80 (4%; CI: 0–11%) wildebeest, 1/25 (4%; CI: 0–20%) zebra and 3/10 (30%; CI: 7–65%) impala whole blood, serum and amniotic fluids samples in Serengeti, Tanzania, by means of *IS711* and *bcsp31* real-time PCRs and AMOS-PCR. These positive occurrences were further characterized as *B. abortus* in lion, buffalo, impala and zebra with a multiplex real-time PCR. The AMOS PCR amplification from this study was faint due to low brucellosis infection, and some of the species-specific bands were not the exact size than the species control and should be interpreted with caution or confirmed by isolation and sequencing. Importantly, the AMOS PCR used in this study cannot identify *B. abortus* biovar 3, a biovar isolated form cattle in Tanzania. Gakuya et al. [29] detected positive *Brucella* spp. reactions in 32/177 (18%; CI: 13–25%) buffaloes, 1/6 (17%; CI: 0–64%) cheetah (*Acynonyx jubatus*), 3/9 eland (30%; CI: 7–70%), 2/5 (40%; CI: 5–85%) elephant, 5/20 (25%; CI: 9–49%) giraffe, 1/4 lion (25%; CI: 0–81%), 1/6 (17%; CI: 0–64%), East African oryx (*Oryx beisa*, close to gemsbok) and 1/8 (13%; CI: 0–53%) warthog serum samples from Kenya by means of *bcsp31* real-time PCR with cycle threshold between 23–36. Successful characterization of *B. melitensis* by AMOS-PCR was obtained in three buffaloes and one giraffe with no image or sequence of product for confirmation. Finally, Katani et al. [41] detected *Brucella* spp. DNA by means of *omp2b* real-time PCR in bushmeat of various

wildlife including African buffalo, bushpig (*Potamochoerus larvatus*), dik-dik (*Madoqua* spp.), eland, hippo, impala, topi (*Damaliscus lunatus*), wildebeest and zebra.

PCR-based diagnostic methods have become widely adopted in the diagnosis of brucellosis due to their high sensitivity and specificity [42–49]. Most PCR assays consistently demonstrate near-perfect analytical and diagnostic specificity, often approaching 100%. PCR targets such as the *IS711* insertion sequence and the *bcsp31* gene are highly specific to *Brucella* spp. and do not cross-react with non-*Brucella* organisms. These findings highlight the robustness of PCR in accurately detecting *Brucella* DNA while minimizing the risk of false positives due to cross-contamination. Furthermore, multiplex PCR offers an efficient and reliable means of simultaneously detecting and differentiating *Brucella* species, with minimal potential for misidentification [50].

PCR assays for brucellosis are mainly applied to *Brucella* culture to detect *Brucella* DNA and identify the *Brucella* species. The studies mentioned above also used PCR to detect Brucella DNA and determine the species in various sample types. These findings indicate that PCR assays are a useful tool for detection, but the incorporation of a confirmation step, such as targeting multiple target genes and sequencing, would greatly enhance diagnostic accuracy.

Overall, the epidemiology and ecology of *Brucella* in wildlife remain neglected. The limited application of molecular tools, the diagnostic constraints of serological tests and the concurrent difficulties in obtaining pure isolates from asymptomatic wildlife carriers hamper our understanding of *Brucella* spp. transmission dynamics, maintenance hosts and zoonotic risks. Clarifying the sylvatic cycle of brucellosis, similar to what has been done for diseases like anthrax and bovine tuberculosis [51,52], is crucial for guiding control strategies and understanding its broader epidemiological significance. This study aimed to establish a comprehensive and evidence-based serological and molecular baseline for *Brucella* spp. in a broad range of South African wild mammals, with particular focus on the characterization of *B. abortus* and *B. melitensis*, the two species of greatest significance in the region. A conservative diagnostic approach was employed, as an animal was considered positive only if it tested positive across multiple assays applied in series, thereby increasing the reliability and specificity of the results.

## Materials and methods

### Ethics statement

This study was performed in accordance with the conditions of the Animal Ethics Committee of the Faculty of Veterinary Science, University of Pretoria (ref nr. REC054–21). Permission to conduct research under Section 20 of the Animal Disease Act 35 of 1984 was granted by the Department of Agriculture, Land Reform and Rural Development under reference nrs. 12/11/1/1955 (HP) for KNP, 12/11/1/1/8 (2058MVA) for LWR, 12/11/1/1 and 12/11/1/1/8 (2399SS) for the other SANParks.

All biological samples were transported to Hans Hoheisen Wildlife Research Station following the guidelines for sample movement previous receipt of Red Cross permits from the State Vet competent for the area of sampling.

### Sample area and collection

Samples were collected over a four-year period from 2021 to 2024 from ecologically diverse locations spanning five of South Africa's nine provinces (Limpopo, Mpumalanga, Northern Cape, Western Cape and Eastern Cape) providing a broad perspective on the brucellosis status in South African wildlife. In fact, 12 nature reserves were included in the study area, comprising six National Parks managed by the South African National Parks (SANParks) node (www.sanparks.org), namely Kruger National Park (KNP), Karoo National Park (KaNP), Camdeboo National Park (CaNP), Mountain Zebra National Park (MZNP), Mokala National Park (MokNP) and Addo Elephant National Park (AENP), as well as 6 private nature reserves, i.e., Lapalala Wilderness Reserve (LWR), Timbavati Private Nature Reserve (TPNR), MalaMala Game Reserve (MMGR), Manyeleti game reserve (MGR), Klaserie private nature reserve (KPNR) and Sabi Sand Nature reserve (SSNR). For data analysis, TPNR, MMGR, MGR, KPNR and SSNR were incorporated with KNP in the Greater KNP

(GKNP) complex since these parks are in continuity and share the same ecosystem and wildlife population. Conversely, GKNP and the other parks exhibit significant variation in terms of ecological, climatic, faunal and managerial characteristics, enhancing the robustness of the study by capturing a wide range of environmental and wildlife conditions. Detailed information about the animals sampled are provided in S1 File.

A total of 722 wild animals belonging to 23 species were sampled opportunistically within the standard park procedures and activities, which included culling operations (conducted for population control or the removal of problem animals) as well as the immobilization of endangered species for purposes such as collar replacement, medical treatment or translocation. No stratified or randomized sampling strategy was applied, as the authors had no direct control over the selection or timing of these park operations. Two sets of samples were collected (Fig 1). The first comprised samples for molecular analyses (n = 585; S1 File), including spleen (when available), liver, pooled lymph nodes from different anatomical districts (submandibular, retropharyngeal, subinguinal, supramammary/supratesticular and tonsillar), EDTA blood and/or serum. Spleen and lymph nodes were analyzed in parallel, while the liver was used as a substitute when spleen samples were unavailable. Where organ samples could not be obtained, as from live animals, only EDTA blood or serum was subjected to molecular testing. The second set comprised samples collected for serological analyses, consisting exclusively of serum (n = 577; S2 File). Not all animals had paired molecular and serological samples available. The number of samples tested stratified per animal species is reported in Fig 1. Organs were chosen due to their role in the reticulo-endothelial and lymphoid systems, where latent infections are reportedly located [53], making them ideal for detecting exposure. Each sample was assigned a unique field identification number and supplementary data were recorded, including the animal species, sex, age, collection date and location. Additional observations, such as the presence of underlying conditions (*e.g.,* presence of macroparasites), clinical signs and, when applicable, necroscopic findings, were also noted. Age was recorded using knowledge of breeding seasons and examination of the physical maturity of the animal to assign it to one of three categories: (i) born in the present breeding season or within the last six months (juvenile); (ii) born the previous breeding season, but not yet mature (sub–adult); (iii) physically mature (adult). GPS coordinates were recorded at the site of sampling, when possible, or at the field abattoir. All sampling equipment and surfaces (knives, chopping boards, camping tables etc…) were rigorously disinfected after processing each individual animal sample using a sequential protocol involving boiling water, quaternary ammonium compound, with non toxic ampholytic surfactants and sequestrants (F10 veterinary disinfectant) and 100% ethanol, followed by thorough physical wiping with disposable paper towels to minimize the risk of cross-contamination. Collected samples were temporarily stored in portable freezers maintained at 4 °C at the field site and subsequently transferred to -20 °C freezers with access control at the Hans Hoheisen Wildlife Research Station, Orpen, Kruger National Park.

## Serological testing

Wildlife sera were first screened using rose Bengal test (RBT) obtained from Onderstepoort Biological Products (OBP, South Africa). RBT was performed by adding 30 µl of rose Bengal reagent and 30 µl of serum. The *Brucella*-positive serum from OBP was used as positive control. Sera were confirmed in series using the Innovative Diagnostics Screen Brucellosis Serum Indirect Multi-species ELISA (iELISA) produced by Innovation Diagnostics (France) to detect antibodies against the lipopolysaccharide (LPS) of smooth *Brucella* spp. The iELISA was performed as per the manufacturer's instructions. Animals were confirmed seropositive only if positive to both RBT and iELISA due to the well documented problem of extensive serological cross-reactions with other bacteria [Godfroid et al., 2012, 54].

## DNA extraction

Each organ sample was prepared by removal of extraneous material (*e.g.,* fat, hairs), cut into small pieces (2 mm$^3$) using sterile forceps and disposable surgical blades. DNA was extracted from the tissue using the PureLink Genomic DNA Kit (Invitrogen, ThermoFisher Scientific, USA), as per the manufacturer's instructions. DNA was extracted from 200 µl of each EDTA blood or serum specimen applying the relevant protocol for "blood lysate".

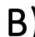

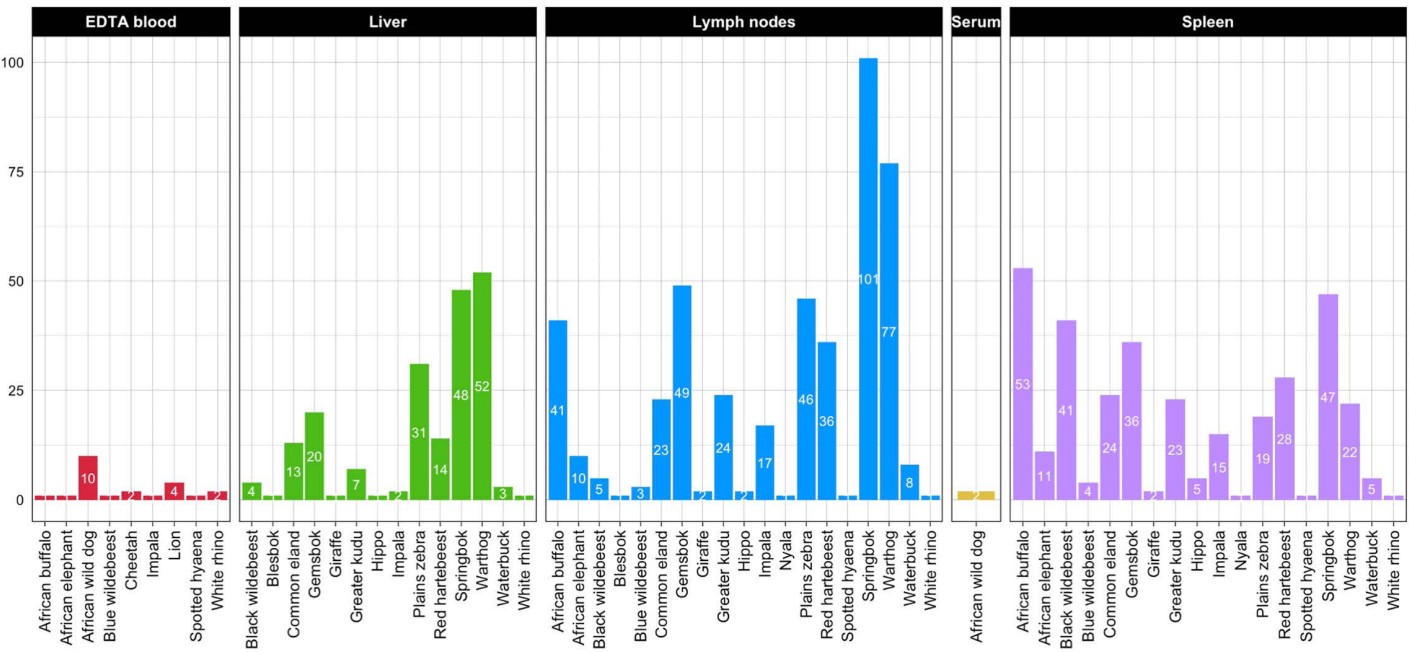

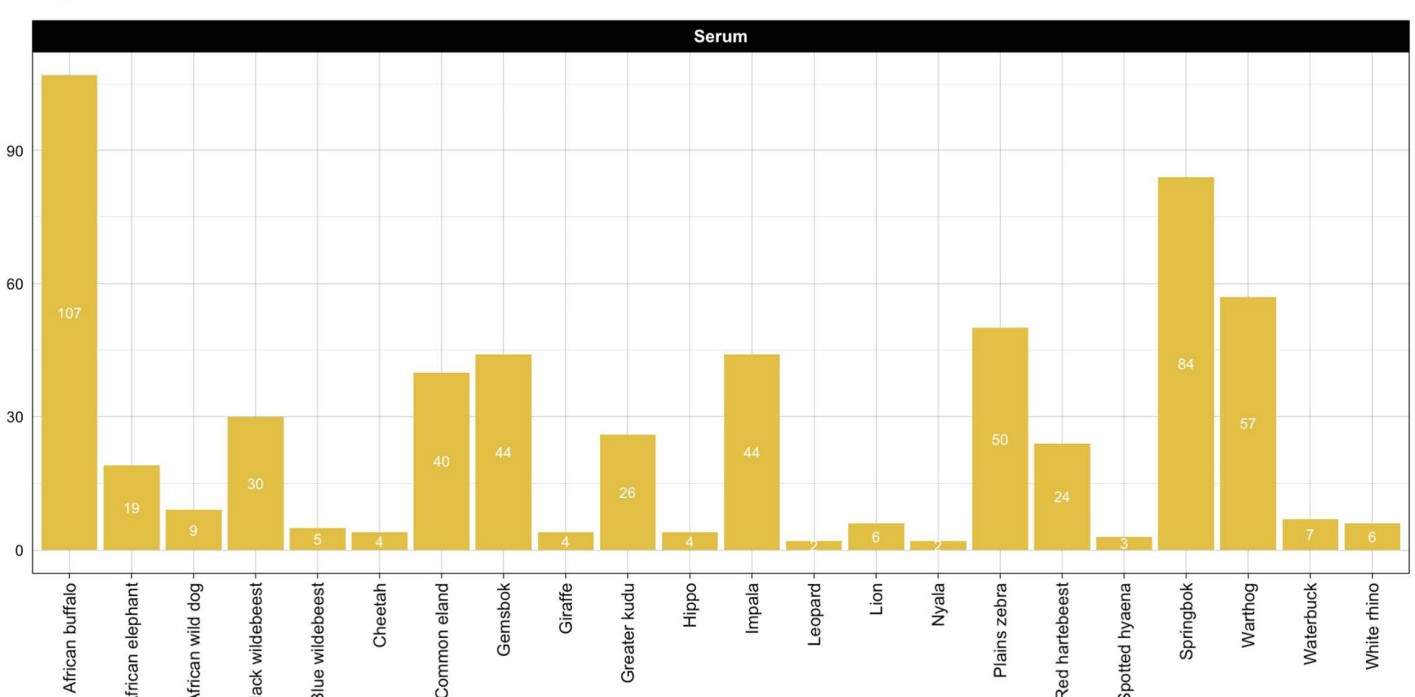

**Fig 1. Barplot representing the frequency distribution of wildlife samples stratified per animal species and sample type.** A) Samples used for molecular testing; B) Samples used for serological testing.

## Molecular testing

Each DNA sample was initially screened for the presence of *Brucella* spp. using a real-time PCR assay targeting *IS711* [55], with the probe labeled with FAM at the 5′ end and BHQ1 at the 3′ end to reduce non-specific amplification. Amplification was performed following the original protocol, and fluorescence was detected using a Bio-Rad CFX Connect Real-Time PCR Detection System. Samples yielding cycle threshold (Ct) values below 35 were considered positive, while those with Ct values between 35 and 37 were classified as suspect. To confirm the presence of *Brucella* DNA at the genus level, all *IS711*-positive and suspect samples were subjected to a conventional PCR (cPCR) targeting the 16S-23S internal transcribed spacer (ITS) region [56]. Only samples that were positive or suspect on *IS711* real-time PCR and confirmed positive by ITS-cPCR were deemed true positives and selected for further species-level characterization.

For species differentiation, a duplex real-time PCR assay [55] was applied to the confirmed samples, again using Ct < 35 as the threshold for positivity and Ct 35–37 as suspect. This reaction was performed on a LightCycler Nano instrument (Roche Diagnostics, Basel, Switzerland). Species identification was further validated using AMOS-PCR [57], conducted in singleplex using species-specific primers for *B. abortus* and *B. melitensis*. As positive controls, DNA from the *B. abortus* RB51 vaccine strain and *B. melitensis* 16M reference strain were included in each run. A detailed description of each PCR assay is reported in S3 File. Given the use of tissue matrices, which often introduce non-specific amplification or contaminating background, touchdown cycles were implemented in both cPCR assays (ITS and AMOS-PCR). These cycling conditions enhanced specificity and consistently produced clean, single bands, eliminating double-band artefacts (S4 and S5 Files).

As required per the Section 20 of the Animal Disease Act 35 of 1984, a subset of samples (a wild dog and a zebra sample from LWR) was sent for confirmation testing with AMOS-PCR at the Agricultural Research Council – Onderstepoort Veterinary Research (ARC-OVI) laboratories.

## DNA sequencing and phylogenetic analysis

A 15% proportion of positive ITS-cPCR products were purified using the QIAquick PCR Purification Kit (Qiagen, Hilden, Germany) according to the manufacturer's protocol. Purified products were submitted to Inqaba (Inqaba Biotechnical Industries, South Africa) for one-direction Sanger sequencing with reverse primer. After quality trimming of sequence ends, secondary peak calling and reverse-complementing sequences, nucleotide data were analyzed for similarity with sequences available in GenBank using the BLAST tool on the NCBI website (http://blast.ncbi.nlm.nih.gov/Blast.cgi). All the sequences obtained in this study were deposited in GenBank database under the accession numbers: PX108534-PX108543. A total of 10 sequences were aligned with selected database reference sequences using MUSCLE algorithm [58]. For each final alignment obtained, nucleotide substitution models were ranked according to the Akaike's Information Criterion (AIC) and Bayesian Information Criterion (BIC) with the "model testing" option of MegaX (Version 10.2.6). After model selection, maximum likelihood and neighbour-joining phylogenetic analyses were performed using MegaX with 1000 bootstrap pseudo–replicates.

## Culture

*Brucella* selective supplement was obtained from ThermoFisher Scientific (Waltham, Massachusetts, USA) to prepare Farrell's media [59]. The antibiotic mix was prepared according to the manufacturer's instructions. Frozen tissues were thawed slowly at fridge temperature (4 °C) and inoculated on selective media. Plates were incubated at 37 °C and 5% $CO_2$ for 7–14 days and examined daily. Suspect colonies were investigated using catalase reaction (degradation of hydrogen peroxide) to identify strict and facultative aerobes, Gram and Stamp's modified Zhiel-Neelsen (MZN) staining [60], and molecular screening using the ITS-cPCR [56]. Colonies confirmed as *Brucella* spp. were further identified using AMOS-PCR [57] and Bruce-Ladder PCR [61].

## Multiloci variable number of tandem repeats

As performed for the *Brucella* isolates from African Buffalo in KNP [24], *Brucella* culture obtained from spleen of plains zebra in LWR was biotyped using MLVA and compared with 115 *Brucella* strains isolated at ARC-OVR from cattle from 1994 and 2006 in South Africa and other strains retrieved from Liu et al. [62] (S6 File) to evaluate their relationships. MLVA-11 and MLVA-16 were performed as previously described [63]. Genotype was scored by visual analysis of the gel images. Band size estimates were converted to repeat units following the published allele calling table [64]. MLVA data were analysed as a character data set within BioNumerics software (version 8.1) (Applied Maths, Sint-Martens-Latem, Belgium). Clustering analysis was performed using the Minimum Spanning Tree method or using the Manhattan distance coefficient and the Bio-Neighbor Joining tree algorithm with permutation resampling as implemented in BioNumerics.

## Data management, analysis and visualization

Sample, laboratory, and storage data were securely registered in a SQLite database via a customized user interface developed specifically for this study. The interface, part of a broader web-based system names "AlephOne", was built using PHP Laravel with Livewire components (https://aleph-one.carlocossu.it/). The application is designed to support field-to-laboratory data integration and real-time data visualisation in one-health disease surveillance, ultimately ensuring traceability across all sample workflows. Data analysis was further performed using the R programming language (version 4.2.1) within the RStudio integrated development environment (IDE) (RStudio Team, 2021). Point prevalence was calculated along with 95% confidence intervals (CIs) to assess variability and precision of the estimates. To estimate CIs, we opted for the conservative Clopper Pearson method [65] using the R function "exactci" from the "PropCIs" package.

To assess if independent variables (*i.e.,* animal family, animal species, sex, age group, sample type and sampling park) significantly influenced infection with a certain pathogen, we employed the Pearson's chi-squared test. Monte Carlo simulation [66] was performed using the option "simulate.p.value = TRUE" in the R function "chisq.test", in order to account for the chi-squared test conditions that were not met (*i.e.*, no cells with expected values <1, and no more than 20% of cells with values <5). We set the number of replicates in the simulation to $B = 2000$ and the statistical level was set at $\alpha = 0.05$.

## Results

The results presented in Table 1 highlight important patterns in the serological and molecular detection of *Brucella* spp. in wildlife. All sampled animals were clinically healthy at the time of collection, without specific signs of brucellosis (*e.g.,* orchitis, arthritis, or abortion). Serological testing detected exposure to *Brucella* spp. in 27/577 (5%) animals (Fig 2A), restricted to 24/106 (23%; CI: 15–32%) African buffalo and 2/4 (50%; CI: 7–93%) giraffes from GKNP, and 1/9 (11%; CI: 0–48%) wild dogs (*Lycaon pictus*) from LWR. Additional 11 buffalo, one giraffe, one hippo and one hyaena from GKNP tested positive only by means of iELISA but negative by RBT, therefore could not be confirmed seropositive. Overall, serological prevalence was low at 5% (30/580; 95% CI: 4–7%), while molecular prevalence was notably higher at 12% (70/585; 95% CI: 9–15%) (see S4 and S5 Files), reaching high prevalences in African buffalo (29/57 = 51%; CI: 37–64%) from GKNP (Fig 2B). Positive cases were also observed in megaherbivores such as 6/15 (40%; CI: 16–68%) elephants, 1/5 (20%; CI: 1–72%) hippo and 1/3 (33%; CI: 1–91%) giraffe from GKNP. A total of 25/53 (47%) animals from LWR tested positive, i.e., 9/17 (53%; CI: 28–77%) impala, 3/12 (25%; CI: 5–57%) wild dogs, 4/4 (100%; CI: 40–100%) zebra, 3/4 (75%; CI: 19–99%) warthogs, 3/3 (100%; CI: 29–100%) blue wildebeest and 1/1 (100%; CI: 3–100%) elephant. Rare occurrences were observed in 2/31 (6%; CI: 1–21%) elands and 1/43 (2%; CI: 0–12%) gemsbok from Karoo National Park, and 3/105 (3%; CI: 1–8%) springboks and 2/55 (4%; CI: 0–13%) warthogs from Mokala National Park. Sequencing and phylogenetic analysis confirmed the specificity of the amplified *Brucella* spp. *ITS* DNA in various wildlife samples (Fig 3). Only 10 animals tested positive on both serological and molecular screening, i.e., nine buffaloes from KNP and one giraffe from TPNR. African buffalo showed the highest prevalence by serological and molecular methods, with a seroprevalence of 22% (CI: 15–32%) and molecular detection of 52% (CI: 38–65%). Several other species, including impala,

**Table 1. Serological and molecular prevalence of *Brucella* spp. in wildlife samples.**

| Animal species | Park | Serological prevalence [95%CI] | Molecular prevalence [95%CI] |
|---|---|---|---|
| African buffalo (*Syncerus caffer*) | Greater Kruger National Park | 24/106 = **23%** [15-32%] | 29/57 = **51%** [37-64%] |
| | Lapalala Wilderness Reserve | 0/1 = 0% [0-98%] | 1/1 = **100%** [3-100%] |
| | Total | 24/107 = **22%** [15-32%] | 30/58 = **52%** [38-65%] |
| African elephant (*Loxodonta africana*) | Greater Kruger National Park | 0/18 = 0% [0-19%] | 6/15 = **40%** [16-68%] |
| | Lapalala Wilderness Reserve | 0/1 = 0% [0-98%] | 1/1 = **100%** [3-100%] |
| | Total | 0/19 = 0% [0-18%] | 7/16 = **44%** [20-70%] |
| African wild dog (*Lycaon pictus*) | Lapalala Wilderness Reserve | 1/9 = **11%** [0-48%] | 3/12 = **25%** [5-57%] |
| Black wildebeest (*Connachaetes gnou*) | Camdeboo National Park | 0/6 = 0% [0-46%] | 0/10 = 0% [0-31%] |
| | Mountain Zebra National Park | 0/24 = 0% [0-14%] | 0/31 = 0% [0-11%] |
| | Total | 0/30 = 0% [0-12%] | 0/41 = 0% [0-9%] |
| Blesbok (*Damaliscus pygargus phillipsi*) | Mokala National Park | Not tested | 0/1 = 0% [0-98%] |
| Blue wildebeest (*Connachaetes taurinus*) | Greater Kruger National Park | 0/2 = 0% [0-84%] | 0/2 = 0% [0-84%] |
| | Lapalala Wilderness Reserve | 0/3 = 0% [0-71%] | 3/3 = **100%** [29-100%] |
| | Total | 0/5 = 0% [0-52%] | 3/5 = **60%** [15-95%] |
| Cheetah (*Acynonyx jubatus*) | Lapalala Wilderness Reserve | 0/4 = 0% [0-60%] | 1/3 = **33%** [1-91%] |
| Common eland (*Taurotragus oryx*) | Camdeboo National Park | 0/6 = 0% [0-46%] | 0/9 = 0% [0-34%] |
| | Karoo National Park | 0/34 = 0% [0-10%] | 2/31 = **6%** [1-21%] |
| | Total | 0/40 = 0% [0-9%] | 2/40 = **5%** [1-17%] |
| Gemsbok (*Oryx gazella*) | Camdeboo National Park | 0/13 = 0% [0-25%] | 0/13 = 0% [0-25%] |
| | Karoo National Park | 0/31 = 0% [0-11%] | 1/43 = **2%** [0-12%] |
| | Total | 0/44 = 0% [0-8%] | 1/56 = **2%** [0-10%] |
| Giraffe (*Giraffa camelopardalis*) | Greater Kruger National Park | 2/4 = **50%** [7-93%] | 1/3 = **33%** [1-91%] |
| Greater kudu (*Tragelaphus strepsiceros*) | Camdeboo National Park | 0/22 = 0% [0-15%] | 0/32 = 0% [0-11%] |
| | Greater Kruger National Park | 0/4 = 0% [0-60%] | 0/1 = 0% [0-98%] |
| | Total | 0/26 = 0% [0-13%] | 0/33 = 0% [0-11%] |
| Hippo (*Hippopotamus amphibius*) | Greater Kruger National Park | 0/4 = 0% [0-60%] | 1/5 = **20%** [1-72%] |
| Impala (*Aepyceros melampus*) | Greater Kruger National Park | 0/29 = 0% [0-12%] | Not tested |
| | Lapalala Wilderness Reserve | 0/15 = 0% [0-22%] | 9/17 = **53%** [28-77%] |
| | Total | 0/44 = 0% [0-8%] | 9/17 = **53%** [28-77%] |
| Leopard (*Panthera pardus*) | Greater Kruger National Park | 0/2 = 0% [0-84%] | Not tested |
| Lion (*Panthera leo*) | Greater Kruger National Park | 0/3 = 0% [0-71%] | Not tested |
| | Lapalala Wilderness Reserve | 0/3 = 0% [0-71%] | 0/4 = 0% [0-60%] |
| | Total | 0/6 = 0% [0-46%] | 0/4 = 0% [0-60%] |
| Nyala (*Tragelaphus angasii*) | Greater Kruger National Park | 0/2 = 0% [0-84%] | 0/1 = 0% [0-98%] |
| Plains zebra (*Equus quagga*) | Addo Elephant National Park | 0/44 = 0% [0-8%] | 0/45 = 0% [0-8%] |
| | Greater Kruger National Park | 0/2 = 0% [0-84%] | 0/1 = 0% [0-98%] |
| | Lapalala Wilderness Reserve | 0/4 = 0% [0-60%] | 4/4 = **100%** [40-100%] |
| | Total | 0/50 = 0% [0-7%] | 4/50 = **8%** [2-19%] |
| Red hartebeest (*Alcelaphus buselaphus*) | Karoo National Park | 0/24 = 0% [0-14%] | 0/44 = 0% [0-8%] |
| Spotted hyaena (*Crocuta crocuta*) | Greater Kruger National Park | 0/2 = 0% [0-84%] | 0/1 = 0% [0-98%] |
| | Lapalala Wilderness Reserve | 0/1 = 0% [0-98%] | 0/1 = 0% [0-98%] |
| | Total | 0/3 = 0% [0-71%] | 0/2 = 0% [0-84%] |
| Springbok (*Antidorcas marsupialis*) | Mokala National Park | 0/84 = 0% [0-4%] | 3/105 = **3%** [1-8%] |

*(Continued)*

**Table 1.** (Continued)

| Animal species | Park | Serological prevalence [95%CI] | Molecular prevalence [95%CI] |
|---|---|---|---|
| Warthog (*Phacocoerus africanus*) | Addo Elephant National Park | 0/14 = 0% [0-23%] | 0/17 = 0% [0-20%] |
| | Greater Kruger National Park | 0/3 = 0% [0-71%] | 0/2 = 0% [0-84%] |
| | Lapalala Wilderness Reserve | 0/4 = 0% [0-60%] | 3/4 = **75%** [19-99%] |
| | Mokala National Park | 0/36 = 0% [0-10%] | 2/55 = **4%** [0-13%] |
| | Total | 0/57 = 0% [0-6%] | 5/78 = **6%** [2-14%] |
| Waterbuck (*Kobus ellipsiprymnus*) | Greater Kruger National Park | 0/1 = 0% [0-98%] | 0/2 = 0% [0-84%] |
| | Mokala National Park | 0/6 = 0% [0-46%] | 0/6 = 0% [0-46%] |
| | Total | 0/7 = 0% [0-41%] | 0/8 = 0% [0-37%] |
| White rhino (*Ceratotherium simum*) | Lapalala Wilderness Reserve | 0/6 = 0% [0-46%] | 0/3 = 0% [0-71%] |
| Overall | | 27/577 = **5%** [3-7%] | 70/585 = **12%** [9-15%] |

plains zebra and blue wildebeest in LWR, had high molecular prevalences (53% [28–77%], 100% [40–100%], and 100% [29–100%], respectively) despite no serological positives. Similarly, elephants and giraffes exhibited molecular evidence of infection (44% [20–70%] and 33% [1–91%], respectively), with elephants showing no serological response. African wild dogs demonstrated both seroprevalence (11%, [0–48%]) and molecular detection (25%, [5–57%]), supporting their potential role in transmission. In contrast, many species, such as black wildebeest, red hartebeest, and greater kudu, showed no evidence of infection by either method.

A total of 21/70 (30%) animals presented multiorgan infection with *Brucella* spp. The majority of *Brucella*-positive sample types (Fig 4A) were obtained from spleen and lymph nodes (49%), followed by spleen alone (24%) and lymph nodes and liver combined (7%). A smaller proportion of detections came from liver (6%), lymph nodes (4%), serum (3%), and EDTA blood (7%). These positive samples were derived from a range of wildlife species and parks. Characterization could be achieved in 50/70 (71%) *Brucella* spp. positive cases, resulting in 26/70 (37%) *B. abortus* and 7/70 (10%) *B. melitensis* cases, with additional 17/70 (24%) mixed *B. abortus-B. melitensis* infections and a substantial portion (29%) remained uncharacterized (Fig 4B). *Brucella abortus* was detected across several species including African buffalo, elephants and springbok, while mixed infections and *B. melitensis* alone were found in species such as impala, blue wildebeest and warthog. *Brucella abortus* positive cultures were obtained from three African buffaloes from KNP and one plains zebra from LWR. Characterization of buffalo cultures has already been reported [24]. In this study, we report the MLVA-16 profiling of an unpure culture obtained from the spleen of a plains zebra from LWR that was identical to one of the African buffalo (strain SAN94), and at one mismatch from the other buffalo isolates from South Africa (strains SAN63 and SAN68), and cattle isolates from Brazil (strains 42–47 and LBAB019 to LBAB047) and Portugal (strains LNIV-416Ba1–07 to LNIV-446Ba1–07).

According to the univariate analysis, "animal family", "animal species", "age group", "park" and "sample type" were significant variables (Table 2). Sample type resulted to significantly influence the detection of *Brucella* spp. in favor of spleen samples, that confirm to be the best sample analyte for this testing approach. Sub-adult and adult Bovidae and Elephantidae were more at risk of infection with *Brucella* spp.

## Discussion

Overall, serological prevalence was low at 5% (27/577) while molecular prevalence was notably higher at 12% (70/585) suggesting that PCR may be more sensitive or capable of detecting infection in the absence of positive serological results (*i.e.,* RBT and ELISA positive). African buffalo showed the highest prevalence by both methods, with a seroprevalence of 22% and molecular detection of 52%, indicating substantial exposure and infection within this species. Several other species, including impala, plains zebra, and blue wildebeest in LWR, had high molecular detection rates (53%, 100%,

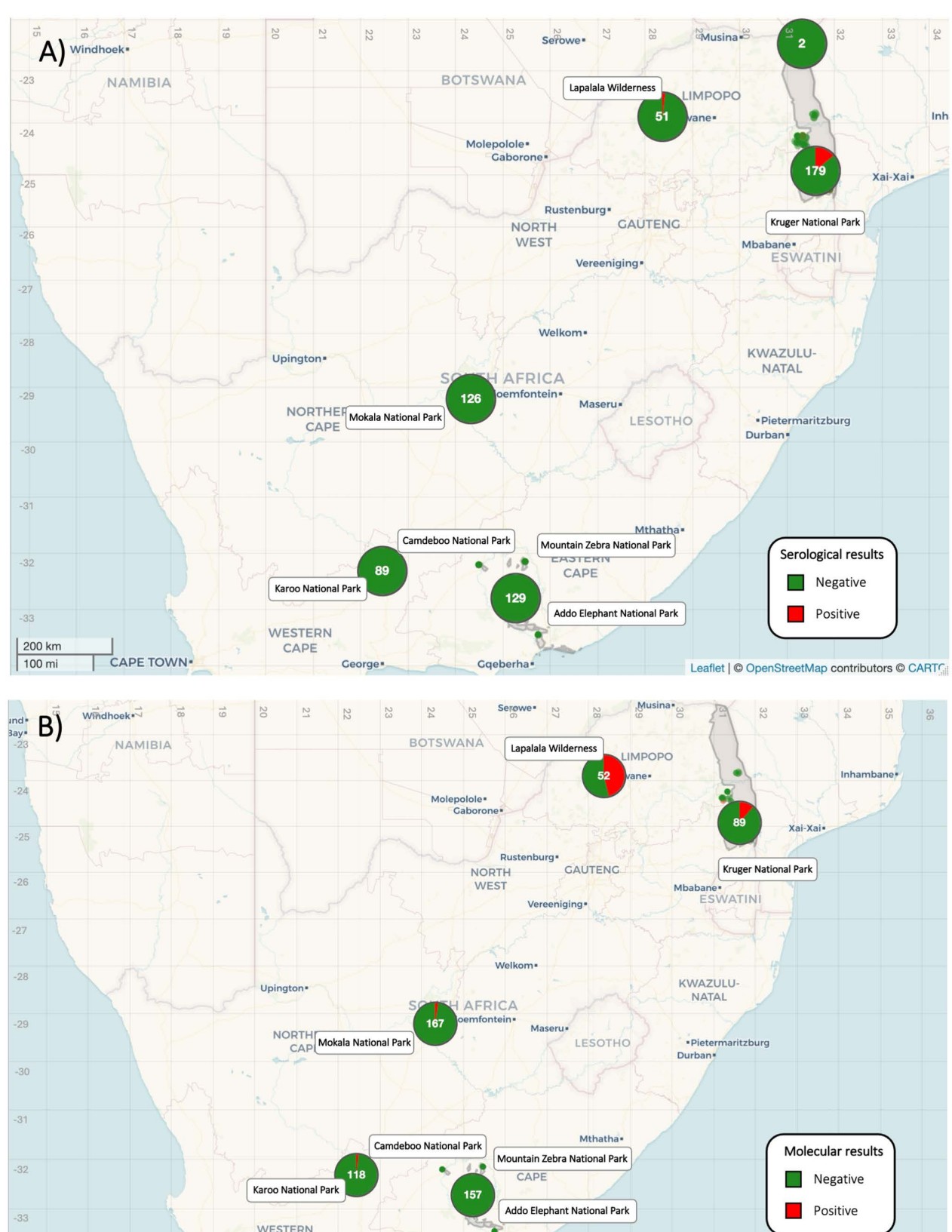

**Fig 2. Geographical distribution of sampled animal locations stratified by test results from (A) serological and (B) molecular analyses.** Circles indicate individual animal locations, while pie-chart markers aggregate results of spatially clustered samples by geographic proximity. Base map source: OpenStreetMap contributors, rendered via CARTO's *CartoDB Voyager* basemap (https://carto.com/attributions), used under the Open Data Commons Open Database License (ODbL 1.0). Park boundary shapefiles were digitized from SANParks administrative data (available upon request from SAN-Parks GIS unit).

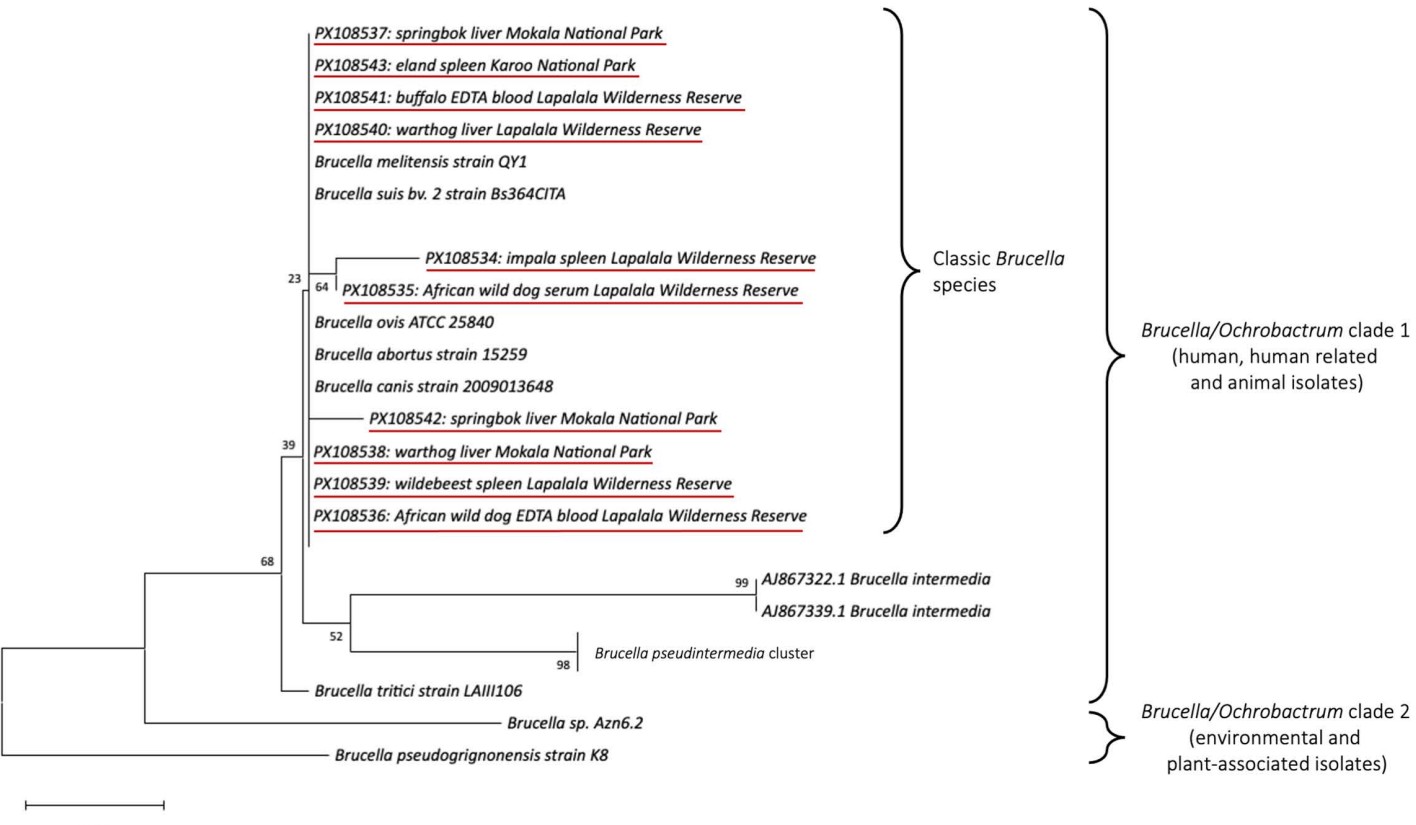

## Neighbour-joining *ITS Brucella* spp. tree

**Fig 3. Neighbour-joining phylogenetic tree displaying the evolutionary relationships of *Brucella/Ochrobactrum* species based on 214 bp of the 23S-5S internal transcribed spacer (*ITS*).** Sequences from this study are underlined in red. Note clear clustering of the sequences obtained with classic *Brucella* spp.

and 100%, respectively) despite no serological positives, pointing to either early infection stages, limitations of serological tests in wildlife or waning antibody levels. Similarly, elephants and giraffes exhibited molecular evidence of infection (44% and 33%, respectively), with elephants showing no serological response (Table 1). African wild dogs demonstrated both seroprevalence (11%) and molecular detection (25%), supporting their potential role in transmission. In contrast, many species, such as black wildebeest, red hartebeest and greater kudu, showed no evidence of infection by either method. These findings underscore the importance of combining molecular and serological tools for accurate detection and surveillance of brucellosis in wildlife populations. Furthermore, the findings suggest the circulation of brucellosis within specific

**A)**

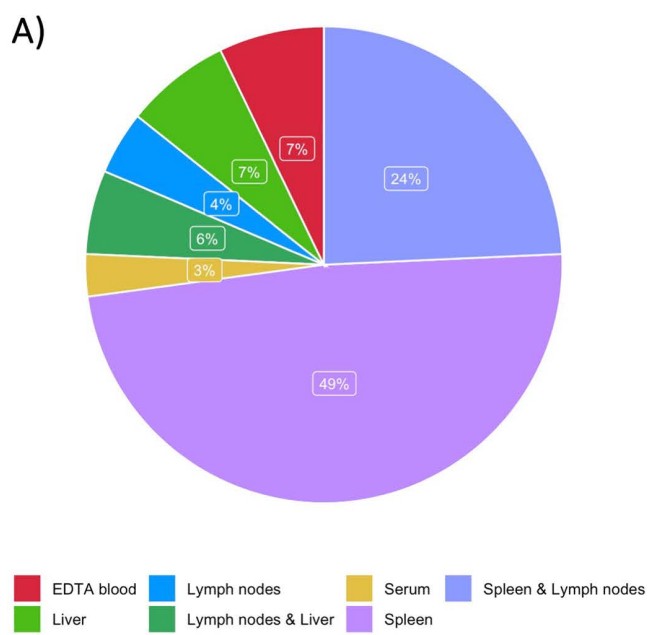

| Legend | |
| --- | --- |
| ■ EDTA blood | ■ Lymph nodes |
| ■ Liver | ■ Lymph nodes & Liver |
| ■ Serum | ■ Spleen & Lymph nodes |
| ■ Spleen | |

| Sample type | Animal species | Park | Count |
| --- | --- | --- | --- |
| EDTA blood | African buffalo | LWR | 1 |
| | African elephant | LWR | 1 |
| | African wild dog | LWR | 1 |
| | Blue wildebeest | LWR | 1 |
| | Cheetah | LWR | 1 |
| Liver | Giraffe | GKNP | 1 |
| | Springbok | MokNP | 2 |
| | Warthog | MokNP | 2 |
| Lymph nodes | Common eland | KaNP | 1 |
| | Impala | LWR | 2 |
| Lymph nodes & Liver | Impala | LWR | 2 |
| | Warthog | LWR | 2 |
| Serum | African wild dog | LWR | 2 |
| Spleen | African buffalo | GKNP | 25 |
| | African elephant | GKNP | 4 |
| | Blue wildebeest | LWR | 1 |
| | Common eland | KaNP | 1 |
| | Gemsbok | KaNP | 1 |
| | Hippo | GKNP | 1 |
| | Springbok | MokNP | 1 |
| Spleen & Lymph nodes | African buffalo | GKNP | 4 |
| | African elephant | GKNP | 2 |
| | Blue wildebeest | LWR | 1 |
| | Impala | LWR | 5 |
| | Plains zebra | LWR | 4 |
| | Warthog | LWR | 1 |

**B)**

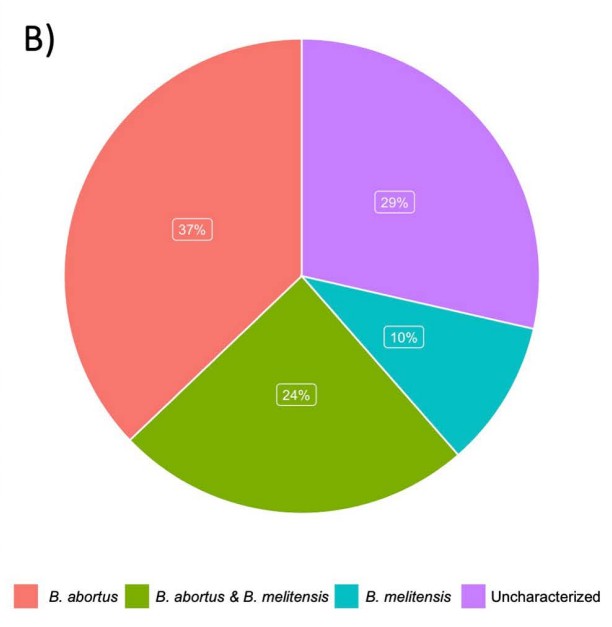

| Legend | |
| --- | --- |
| ■ *B. abortus* | ■ *B. abortus & B. melitensis* |
| ■ *B. melitensis* | ■ Uncharacterized |

| Pathogen species | Animal species | Park | Count |
| --- | --- | --- | --- |
| *B. abortus* | African buffalo | GKNP | 15 |
| | | LWR | 1 |
| | African elephant | GKNP | 2 |
| | Blue wildebeest | LWR | 2 |
| | Gemsbok | KaNP | 1 |
| | Impala | LWR | 3 |
| | Springbok | MokNP | 1 |
| | Warthog | LWR | 1 |
| *B. abortus & B. melitensis* | African buffalo | GKNP | 5 |
| | African elephant | GKNP | 1 |
| | African wild dog | LWR | 3 |
| | Blue wildebeest | LWR | 1 |
| | Impala | LWR | 3 |
| | Plains zebra | LWR | 3 |
| | Warthog | LWR | 1 |
| *B. melitensis* | African elephant | GKNP | 3 |
| | Common eland | KaNP | 1 |
| | Hippo | GKNP | 1 |
| | Impala | LWR | 1 |
| | Plains zebra | LWR | 1 |
| Uncharacterized | African buffalo | GKNP | 9 |
| | African elephant | LWR | 1 |
| | Cheetah | LWR | 1 |
| | Common eland | KaNP | 1 |
| | Giraffe | GKNP | 1 |
| | Impala | LWR | 2 |
| | Springbok | MokNP | 2 |
| | Warthog | MokNP | 2 |
| | | LWR | 1 |

**Fig 4. Pie charts and descriptive tables representing the distribution of the 70 *Brucella* spp. positive animals according to A) infected sample types, and B) identified *Brucella* species.** Note the occurrence of multiorgan infections and *B. abortus* and *B. melitensis* co-infections. GKNP = Greater Kruger National Park; LWR = Lapalala Wilderness Reserve; MokNP = Mokala National Park; KaNP = Karoo National Park.

**Table 2. Results of Pearson's chi–squared test with Monte Carlo simulation where prevalence has been used as outcome variable.**

| Variable | p-value ($X^2$ values) | Values | Positive/Tested = Prevalence [95% CI] |
|---|---|---|---|
| Animal family | < 0.001 (36.9) | *Elephantidae* | 7/16 = **44%** [20-70%] |
| | | *Bovidae* | 48/409 = **12%** [9-15%] |
| | | *Giraffidae* | 1/3 = **33%** [1-91%] |
| | | *Hippopotamidae* | 1/5 = **20%** [1-72%] |
| | | *Equidae* | 4/50 = **8%** [2-19%] |
| | | *Suidae* | 5/78 = **6%** [2-14%] |
| | | *Canidae* | 3/12 = **25%** [5-57%] |
| | | *Rhinocerotidae* | 0/3 = 0% [0-71%] |
| | | *Felidae* | 1/7 = **14%** [0-58%] |
| | | *Hyaenidae* | 0/2 = 0% [0-84%] |
| Age group | < 0.001 (26.1) | Adult | 25/252 = **10%** [7-14%] |
| | | Sub-adult | 40/268 = **15%** [11-20%] |
| | | Juvenile | 4/62 = **6%** [2-16%] |
| Park | < 0.001 (442.8) | Greater Kruger National Park | 37/90 = **41%** [31-52%] |
| | | Mountain Zebra National Park | 0/31 = 0% [0-11%] |
| | | Karoo National Park | 3/118 = **3%** [1-7%] |
| | | Mokala National Park | 5/167 = **3%** [1-7%] |
| | | Camdeboo National Park | 0/64 = 0% [0-6%] |
| | | Addo Elephant National Park | 0/62 = 0% [0-6%] |
| | | Lapalala Wilderness Reserve | 25/53 = **47%** [33-61%] |
| Sample type | < 0.001 (99.1) | Liver | 9/198 = **5%** [2-8%] |
| | | Spleen | 34/125 = **27%** [20-36%] |
| | | Lymph nodes | 20/228 = **9%** [5-13%] |
| | | EDTA blood | 5/32 = **16%** [5-33%] |
| | | Serum | 2/2 = **100%** [16-100%] |
| Sex | 0.872 (0) | Male | 37/360 = **10%** [7-14%] |
| | | Female | 33/224 = **15%** [10-20%] |

South African wildlife populations, highlighting the limitations of relying solely on serological surveillance. The ecological role of each animal species in the sylvatic cycle of *Brucella* spp. remains largely unclear and must be interpreted in light of local epidemiological conditions, host ecology and patterns of interspecies contact.

African Buffalo in the GKNP exhibited high seroprevalence (24/106; 23%, CI: 15–32%) and even higher molecular prevalence (29/57; 51%, CI: 37–64%), significantly exceeding previous serological estimates [26], as well as molecular prevalences documented in the Serengeti ecosystem, Tanzania [40], and Kenya [29]. This provides strong evidence of *Brucella* spp. being endemically established in GKNP, with African buffalo likely serving as the main wildlife reservoir for *B. abortus* [24,67]. The detection of *B. melitensis* in co-infection with *B. abortus* in some buffaloes, that is seldom reported in livestock species, further underscores the complexity of brucellosis in wildlife, raising questions about the mechanisms enabling co-infection, potential transmission routes and the concurrent circulation of multiple *Brucella* species within the same host population.

Unexpected *Brucella* detections in non-bovid species such as zebras, elephants, hippopotamuses and giraffes further challenge assumptions about host specificity/preference. Although these animal species have not traditionally been considered core hosts for *Brucella* spp., they may participate in transmission dynamics or even pathogen maintenance under certain ecological conditions. Elephants and hippopotamuses, for instance, often graze in areas frequented by ruminants like buffalo, potentially exposing them to contaminated pastures or water sources [68,69]. The detection of *B. melitensis* in elephants and hippopotamus is especially notable, given that this *Brucella* species is primarily associated with sheep and goats, suggesting livestock-independent wildlife cycles. One giraffe was confirmed positive by both serological and molecular methods, providing dual confirmation of infection in this species. Giraffes, though primarily browsers, are known to exhibit pica-related behaviors such as osteophagia and geophagia, particularly during periods of nutritional stress or drought [70], which may facilitate ingestion of infected tissues or materials, including aborted fetuses.

In LWR, molecular prevalence was similarly high, with 25/53 (47%) animals (47%) from seven species testing positive. Notably, 11 individuals were co-infected with both *B. abortus* and *B. melitensis*. Carnivores such as African wild dog and cheetah also tested positive, suggesting the potential for multi-host transmission, possibly through predation or scavenging of infected prey [17]. One impure culture of *B. abortus* was isolated from the spleen of a plains zebra. MLVA revealed genetic clustering with South African buffalo isolates and, interestingly, with South American cattle strains, raising questions about *B. abortus* strain movement through trade, translocation or shared environmental sources.

Additional *Brucella* detections included *B. melitensis* in eland and *B. abortus* in gemsbok from Karoo National Park, as well as *Brucella* spp. DNA in warthog and *B. abortus* in springbok from Mokala National Park. These incidental findings may represent spillover infections with limited epidemiological relevance for local transmission or potentially reflect dormant infections that do not contribute to sustained transmission cycles. However, these findings require careful consideration, as wildlife species once thought to play minimal roles may become more prominent under shifting ecological or climatic conditions. For example, environmental stressors or changes in species distribution could alter interspecies contact rates, eventually reshaping transmission dynamics [71].

Serological testing remains one of the primary methods for brucellosis surveillance in wildlife, although it has notable limitations. Most serological assays have been developed and validated for use in livestock and their diagnostic performance in wildlife species is not well established. Even so-called "multi-species" indirect ELISAs, developed for domestic animals, may not perform optimally when applied to wildlife, as differences in immunoglobulin structure across species can affect antibody detection. For instance, studies using different conjugates, including proteins A, G and A/G [72–74], have demonstrated suboptimal performance in wildlife species, highlighting the incompatibility of certain reagents with non-domesticated hosts. Protein A/G indirect ELISA was shown to be optimal for the detection of anti-Brucella antibodies in Arctic wildlife [73]. Although the ID Screen multispecies Brucellosis indirect ELISA employs a non-species-specific IgG-HRP conjugate, the precise cross-reactivity and sensitivity of this conjugate across diverse wildlife taxa remain unclear [23]. This raises concerns that it may suffer from similar limitations as other assays tested with non-validated conjugates. To improve diagnostic reliability in wildlife, it is essential to use appropriately validated reagents. Recent advances, such as the development of wildlife-specific conjugates by [75], provide a promising step forward. Incorporating wildlife-specific conjugates into indirect or competitive ELISAs could significantly enhance the sensitivity and specificity of serological detection in a range of wild species, and should be considered in future surveillance efforts. In wildlife, exposure to *Brucella* spp. is likely to be detected as a chronic infection. In chronic infections in livestock, the weaning serological antibodies may not be detected anymore by the RBT whereas they remain detectable by iELISA [76]. This suggests that our definition of positivity in wildlife may be too conservative and that, when tested in parallel, positivity with either RBT or ELISA could be considered sufficient.

In this study, molecular detection consistently outperformed serological testing across wildlife species, as evidenced by a higher overall PCR prevalence (12%) compared to serology (5%). This finding highlights the value of molecular tools, particularly PCR, for the sensitive detection of *Brucella* DNA in wildlife. Unlike previous studies that relied primarily on

whole blood or serum for molecular testing [29,41,40], our results demonstrated that tissue samples, especially spleen and lymph nodes, yielded a greater number of positive detections, whereas whole blood accounted for only a small proportion of positives (7%). These findings suggest that tissue samples may provide a higher diagnostic sensitivity for brucellosis in wildlife. However, obtaining tissue samples from free-ranging wildlife is often logistically challenging and not feasible for routine surveillance or diagnostic purposes. In contrast, whole blood remains a more accessible and less invasive sample type [77]. Importantly, in some instances in this study, molecular testing of whole blood and/or serum yielded positive results, indicating that whole blood may still serve as a suitable sample type for molecular diagnosis under field conditions. Given its practicality and reasonable performance, especially when paired with sensitive PCR assays, blood may offer an ideal compromise for wildlife brucellosis surveillance when tissue samples are not available.

A major limitation of this study and wildlife brucellosis research more broadly, is the difficulty in isolating viable *Brucella* spp. from field samples, particularly from animals with chronic or latent infections where bacterial loads are low [53]. Only three pure cultures from buffalo [24] and one impure culture from zebra were identified as *B. abortus* with PCR. The lack of pure cultures limited our ability to conduct comperehensive downstream analyses, including whole genome sequencing and phenotypic characterization. The impure *B. abortus* culture recovered from a plains zebra was used in MLVA. However, the results should be interpreted with caution due to mixed bacterial populations. Despite this limitation, the MLVA profile of the zebra-derived isolate showed genotypic linkage consistent with that of pure *B. abortus* isolates obtained from African buffalo, suggesting a potential epidemiological connection between infections in these two species. Furthermore, small sample sizes for some animal species prevent reliable estimation of prevalence or confirmation of their epidemiological significance.

Future research should prioritize expanded molecular surveillance of brucellosis in wildlife in areas where *Brucella* spp. have been consistently detected, particularly in the GKNP and LWR. A more detailed understanding of infection prevalence, host range and strain diversity within these ecosystems is essential to refine the sylvatic epidemiological baseline, identify the maintenance host community and distinguish endemic persistence from recent spillover events. Equally important is the inclusion of rural communities living at the interface of protected areas. These communities often depend heavily on animal resources and are at high risk of zoonotic exposure through direct contact with wildlife and livestock. Protected areas are not ecologically isolated. Digging species (*e.g.,* warthogs, hyenas, wild dogs), antelopes, elephants and lions frequently move across reserve boundaries [78,79], creating potential corridors for pathogen dissemination. A One Health approach is key to mitigating the impact of *Brucella* spp. on biodiversity and agricultural productivity but also essential for developing effective, evidence-based disease control strategies at the wildlife interface.

## Supporting information

**S1 File. Comprehensive dataset containing detailed information on animal samples used for molecular testing of *Brucella* spp.**
(XLSX)

**S2 File. Comprehensive dataset containing detailed information on animal sera used for serological testing of *Brucella* spp.**
(XLSX)

**S3 File. Detailed description of PCR assays used in this study.**
(ZIP)

**S4 File. Graphical output of the molecular screening for *Brucella* spp.** On top, typical amplification curves obtained with real-time PCR targeting the insertion sequence 711 (*IS711*), unique to *Brucella* genus. Positive cutoff indicated in

red, "suspect" cutoff indicated in yellow. At the bottom, gel picture of 214 bp PCR products of the *16S-23S ITS* region of *Brucella* spp., reloaded in series to verify correct band size. Red arrows indicate positive reactions.
(PNG)

**S5 File. Graphical output of the molecular characterization for A)** ***Brucella abortus*** **and B)** ***B. melitensis*.** On top, typical amplification curves obtained with duplex real-time PCR. Positive cutoffs are indicated in black, "suspect" cutoffs indicated in grey. At the bottom, gel picture of some AMOS singleplex PCR products where *B. abortus* positive reactions are seen as 500 bp bands and *B. melitensis* as 700 bp bands. Products were here reloaded in series to double-check correct band size.
(PNG)

**S6 File. Strains retrieved from Liu et al. [62] for MLVA typing.**
(XLSX)

## Acknowledgments

The authors acknowledge the hard work and dedication of the veterinary students that aided in sample collection: S. Anders, S. Cuthbert, J. Vaughan, D. Fletcher, J. Lähdeaho, L. Mashanda, M. P. Segopolo, X. Juhl-Jürgens, I. Kruger, J. Kruger, E. Mearns, J. Scheepers, Xanthe, E. S. Day, E. Hosking, F. Perencin, F. Tenani, T. Pettenazzo, C. Fourie, D. Beling, L. Nedz, C. Robertson, C. Bronkhorst, N. Lonsdale, J. van Vuuren. We extend our sincere gratitude to Schalk van Schalkwyk and Rudi Lorist for their invaluable assistance with sample collection in Kruger National Park, as well as Paul White for assistance in Timbavati Private Nature Reserve. Additionally, we thank Dr. Alischa Henning for her efforts in recruiting students to support the sample collection process.

## Author contributions

**Conceptualization:** Carlo Andrea Cossu, Henriette van Heerden.

**Data curation:** Carlo Andrea Cossu, Giuliano Garofolo, Henriette van Heerden.

**Formal analysis:** Carlo Andrea Cossu, Jeanette Wentzel, Giuliano Garofolo, Henriette van Heerden.

**Funding acquisition:** Henriette van Heerden.

**Investigation:** Carlo Andrea Cossu, Jeanette Wentzel, Lin-Mari de Klerk, Jacques Godfroid, Giuliano Garofolo, Henriette van Heerden.

**Methodology:** Carlo Andrea Cossu, Henriette van Heerden.

**Project administration:** Henriette van Heerden.

**Resources:** Henriette van Heerden.

**Software:** Carlo Andrea Cossu.

**Supervision:** Henriette van Heerden.

**Validation:** Carlo Andrea Cossu, Jeanette Wentzel, Lin-Mari de Klerk, Fabrizio De Massis, Jacques Godfroid, Louis Ockert van Schalkwyk, Giuliano Garofolo, Henriette van Heerden.

**Visualization:** Carlo Andrea Cossu.

**Writing – original draft:** Carlo Andrea Cossu.

**Writing – review & editing:** Carlo Andrea Cossu, Jeanette Wentzel, Lin-Mari de Klerk, Fabrizio De Massis, Jacques Godfroid, Louis Ockert van Schalkwyk, Giuliano Garofolo, Henriette van Heerden.

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
