## [Decision Letter · Decision Letter 0]

2 Oct 2025

Epidemiological baseline of Brucella spp. in South African wildlife

Dear Dr. Cossu,

Thank you for submitting your manuscript to PLOS Neglected Tropical Diseases. After careful consideration, we feel that it has merit but does not fully meet PLOS Neglected Tropical Diseases's publication criteria as it currently stands. Therefore, we invite you to submit a revised version of the manuscript that addresses the points raised during the review process.

Please submit your revised manuscript within 60 days Dec 01 2025 11:59PM. If you will need more time than this to complete your revisions, please reply to this message or contact the journal office at plosntds@plos.org. Please include the following items when submitting your revised manuscript:

We look forward to receiving your revised manuscript.

Kind regards,

Javier Pizarro-Cerda

Academic Editor

Stuart Blacksell

Section Editor

Shaden Kamhawi

co-Editor-in-Chief

Paul Brindley

co-Editor-in-Chief

**Additional Editor Comments:**

The study is a valuable analysis of the presence of Brucella spp. in diverse animal wild life species in South Africa. However, the granularity of the study lacks critical information, including sampling strategy, period of data collection, health status of the animals or the origin of the different Brucella species identified in this work, among several missing parameters. The authors should precise these variables in order to truly allow to understand the persistence of Brucella in the studied ecosystem.

**Journal Requirements:**

1) Please provide an Author Summary. This should appear in your manuscript between the Abstract (if applicable) and the Introduction, and should be 150-200 words long. The aim should be to make your findings accessible to a wide audience that includes both scientists and non-scientists. Sample summaries can be found on our website under Submission Guidelines:

- ® on page: 10 and 22.

4) Please ensure that the funders and grant numbers match between the Financial Disclosure field and the Funding Information tab in your submission form. Note that the funders must be provided in the same order in both places as well.

**Reviewers' Comments:**

Reviewer's Responses to Questions

**Key Review Criteria Required for Acceptance?**

**Methods:**

-Are the objectives of the study clearly articulated with a clear testable hypothesis stated?

-Is the study design appropriate to address the stated objectives?

-Is the population clearly described and appropriate for the hypothesis being tested?

-Is the sample size sufficient to ensure adequate power to address the hypothesis being tested?

-Were correct statistical analysis used to support conclusions?

-Are there concerns about ethical or regulatory requirements being met?

Reviewer #1: The objectives of the study are clearly articulated.

The study design is appropriate to address the stated objectives.

However, the samples are not clearly described: the selection of the animals included in the study is not well documented. There is no supporting file listing the samples and their associated data. We don’t know when the samples were collected.

This lack of information makes it challenging to evaluate the study's methodology and reproducibility.

Reviewer #2: Which chemistry was used for real-time PCR experiments? SYBR Green/probes?. If SYBR green was used, melting curves would be necessary for specificity analysis.

Line 2111: All the sequences obtained in this study were deposited in GenBank database under the accession numbers: …

Please indicate

Supplementary figures would benefit from clarification. What do the red arrows indicate? It would be helpful to show in the graphic the size of the correct bands for B. abortus and B.melitensis

Reviewer #3: The authors have carried out extensive sampling work on numerous large wild animals. The article is clear and understandable.

This article is the result of international collaboration. The subject is important for detection and identification, particularly in the context of veterinary and human diagnosis. Numerous publications such as Qureshi et al., 2023, Middlebrook et al., 2022, Gakuya et al., 2022 (cited by the authors) and Dadar et al., 2021 describe the presence of Brucella spp. in wildlife and livestock in sub-Saharan Africa.

The originality of this article lies in the fact that it describes for the first time a very detailed epidemiology of the Brucella genus based on samples taken from more than 500 wild mammals, representing more than 20 different species, in protected areas in South Africa.

In the introduction, could you specify that Brucella is a highly pathogenic bacterium. Samples and isolates must be handled in a biosafety level 3 laboratory in order to protect the handler.

Lines 39 to 41: Could you please specify the origin of the different Brucella species? For example, B. inopinata was isolated from human samples, B. microti was isolated from voles, and B. ceti was isolated from aquatic animals.

You mention the genus Ochrobactrum. Could you clarify that since 2020 (Hördt et al.), certain species of Ochrobactrum (O.) have become Brucella, such as O. anthropi, etc.

Line 138: it is noted ‘A total of 588 wild animals belonging to 22 species were sampled ...’. Table 1 lists 580 animals belonging to 23 species. Could you please correct this information ?

Line 212: It would appear that the number of sequences aligned with the MUSCLE algorithm is missing.

Line 220 : The bibliographic reference for Farrell's medium preparation is missing.

The real-time PCR protocols are clear and combine the various PCRs needed to detect and identify the Brucella genus and its different species.

However, for the ‘Brucella spp. IS711 real-time PCR’ protocol, “BHQ1” is listed for the 3'Quencher, whereas in the article, the 3'Quencher is ‘TAMRA’. Could you please correct this ?

In the reference to Keid et al., the year is listed as 1997, whereas the article was published in 2007. Could you please correct the date in the text of the article, in the bibliographical references and in the ‘Brucella spp. ITS Touchdown cPCR’ protocol ?

**Results**

-Does the analysis presented match the analysis plan?

-Are the results clearly and completely presented?

-Are the figures (Tables, Images) of sufficient quality for clarity?

Reviewer #1: The results are clearly presented. The figures and tables are clear.

However, the serological prevalence must be checked as it is 30/577 (line 258), 30/580 in table 1 and I count 27/577.

Reviewer #2: (No Response)

Reviewer #3: Line 258 : It is noted 30/580 in Table 1. Could you correct either of the values?

Line 263 : The values 30/580 are noted in the text and in Table 1. Are these the correct values ?

Line 267 : « A total of 25/53 (47%) animals from LWR tested positive ». It should be noted that the small number of animals for certain species should be taken into account, with only one animal tested for African buffalo (1/1=100%), African elephant and cheetah (1/3=33%).

The tables help to understand the text. However, a map of South Africa showing the locations of the various animal parks would be greatly appreciated by readers. On this map, you could also show where the largest number of Brucella spp. positive animals are located.

Figures S3 and S4 lack sharpness. Could you please improve the sharpness of the photos ?

**Conclusions:**

-Are the conclusions supported by the data presented?

-Are the limitations of analysis clearly described?

-Do the authors discuss how these data can be helpful to advance our understanding of the topic under study?

-Is public health relevance addressed?

Reviewer #1: The conclusions are well supported by the data.

The limitations of analysis are clearly described.

The authors discuss well. However, without information on when the samples were collected, it is challenging to analyze the long-term persistence of Brucella in the animals over the years.

Reviewer #2: (No Response)

Reviewer #3: In the discussion, the reader clearly understands the fundamental role of monitoring Brucella spp. from asymptomatic wildlife and the role of climate change.

The authors show that certain animals contribute to the dynamics of pathogen transmission without being carriers of these pathogens.

The authors also mention the limitations of their study, in particular the viable isolation of Brucella from environmental or veterinary samples.

Among all the animals you studied, could you please clearly specify the number of healthy carriers and the number of animals showing symptoms ?

Lines 326 to 327 : Specify the number of animals tested for each positive result (53% (9/17), 100% (4/4), 100% (3/3).

Line 335 : ‘the circualtion’ should be corrected to ‘the circulation’

**Editorial and Data Presentation Modifications?**

Reviewer #1: Abstract

Line 25: Please, remove « was successfully » to make an understandable sentence or rewrite the sentence.

Introduction

Lines 78 to 82: This sentence is too long. Please, cut it and indicate the reference number of Sambu et al. (2021) from the list of references. I think it is the 38th.

Lines 86 to 92: This sentence is too long. Please, cut it and indicate the reference number of Gakuya et al. (2022) from the list of references. I think it is the 32nd.

Line 92: indicate the reference number of Katani et al. (2021) from the list of references. I think it is the 39th.

Line 115. Please, insert a short sentence to explain who conceived and designed the study.

Materials and methods

Sample area and collection.

Line 126: Please remove “Fig.1” as this figure does not describe the parks.

Line 138: the animals were sampled opportunistically. Could you precise the period of collect (several years?), the number of live and dead animals. How were the animals selected?

Line 147 to 153: Where are these supplementary data? Please provide a file with these data.

Serological testing

Lines 174 to 176: RBT and ELISA detect the same antigen and display the same cross-reactions with other bacteria. So, using both RBT and ELISA does not improve the specificity, but probably improves the sensitivity of the serological testing. Could you change the sentence?

Line 176: Please indicate the reference numbers

Molecular testing

Line 190: Please indicate the reference numbers

DNA Sequencing

Line 212: the accession numbers are missing

MLVA

What does mean ARC-OVR?

Line 231: Please, correct retrieved from Liu et al (59) (supporting file S4)

Line 233: Maybe you forgot “from” before Vergnaud et al.

Results

Line 258: It seems that the serological prevalence is 27/577. It is not concordant with the results in line 258 (30/577) neither with the overall in table 1(30/580). Please, indicate the correct results.

Figure 2 legend. Line 288: write species instead of specues.

Line 299: you could precise Fig. 3B instead of Fig. 3.

Line 313: Please, make the raw data available for each animal in an additional supporting file: animal species, age group, park, sample type, date of sampling, condition dead or alive, serological and molecular tests.

Lines 303 to 306: The MLVA-16 profiling of an unpure culture obtained from the spleen of a plains zebra was identical to that of the strain SAN94. This strain is listed in the supporting Excel file. However, I cannot find the other strains (Bru 0834, 0874….) in the file. Please, could you check or explain why?

Discussion

Line 344: African buffalo showed a high prevalence in GKNP. You assume that Brucella spp. is endemically established in GKNP. However, this high prevalence could result from an outbreak. Could you discuss it?

Line 357: A giraffe was confirmed positive by both serological and molecular methods. You were able to withdraw some blood and take a piece of liver on this animal? Was this giraffe sick?

Line 392: remove “by”

Line 419: Replace out ability by our ability.

Supporting files

The supporting Information S2 is an xlsx file and correspond to the S4 file listed line 477.

Please check the name of the files.

Add a supporting file with the samples and their associated data.

Reviewer #2: Line 16: Organ and serum samples from (n=588, animals representing 23 species) vs Line 138: A total of 588 wild animals belonging to 22 species

Please homogenize

Line 287. Figure 2. Neighbour-joining phylogenetic tree displaying the evolutionary relationships of Brucella/Ochrobactrum specues

Species

Line 336 Furthermore the findings suggest the circualtion of brucellosis within specific South African wildlife populations, highlighting the limitations of relying solely on serological surveillance

Circulation

Reviewer #3: (No Response)

**Summary and General Comments**

Reviewer #1: This study is notable for demonstrating the high prevalence of Brucella in African buffalo within the vast Kruger National Park, and for highlighting the superior sensitivity of PCR over serological tests in detecting Brucella infections in animals.

However, I have a significant concern regarding this manuscript. The selection of the animals included in the study is not well documented. There is no supporting file listing the samples and their associated data. We don’t know when the samples were collected.

This lack of information makes it challenging to evaluate the study's methodology and the conclusions.

Reviewer #2: I read with interest the manuscript by Cossu et al. The ecological role of wild animal species in the sylvatic cycle of Brucella spp. remains unclear and needs further investigation. In the present study, the authors analyzed brucellosis status of wild animals from five of South Africa's nine provinces (Limpopo, Mpumalanga, Northern Cape, Western Cape, and Eastern Cape), providing a broad perspective on the brucellosis status in wildlife (a total of 588 wild animals belonging to 22 species). Diagnostic procedures combined serology (rose bengal test confirmed by indirect ELISA) and PCR-based assays. The paper is well written and interesting. Co-infections with B. abortus and B. melitensis were identified. Serological prevalence was low at 5% (30/580) while molecular prevalence was notably higher at 12% (70/585), suggesting that PCR may be more sensitive or capable of detecting infection in the absence of positive serological results. The present manuscript underscores the importance of combining molecular and serological tools for accurate detection and surveillance, highlighting the limitations of relying solely on serological surveillance.

Reviewer #3: (No Response)

PLOS authors have the option to publish the peer review history of their article (what does this mean? ). If published, this will include your full peer review and any attached files.

**Do you want your identity to be public for this peer review?** For information about this choice, including consent withdrawal, please see our Privacy Policy .

Reviewer #1: No

Reviewer #2: No

Reviewer #3: No

**Figure resubmission:**
---

## [Decision Letter · Decision Letter 1]

30 Oct 2025

Response to Reviewers
Revised Manuscript with Track Changes
Manuscript

We look forward to receiving your revised manuscript.

Kind regards,

Javier Pizarro-Cerda

Academic Editor

Stuart Blacksell

Section Editor

Shaden Kamhawi

co-Editor-in-Chief

Paul Brindley

co-Editor-in-Chief

**Additional Editor Comments:**

As suggested by Reviewer 3, please include a figure showing sample locations and number of animal studies on a map of South Africa, which would allow to better illustrate the study

**Journal Requirements:**

**Reviewers' comments:**

Reviewer's Responses to Questions

**Key Review Criteria Required for Acceptance?**

**Methods**

-Are the objectives of the study clearly articulated with a clear testable hypothesis stated?

-Is the study design appropriate to address the stated objectives?

-Is the population clearly described and appropriate for the hypothesis being tested?

-Is the sample size sufficient to ensure adequate power to address the hypothesis being tested?

-Were correct statistical analysis used to support conclusions?

-Are there concerns about ethical or regulatory requirements being met?

Reviewer #1: The objectives are clearly articulated.

The study design is appropriate

The population is clearly described

Reviewer #2: The revised version of the manuscript shows significant improvement and adequately addresses the reviewers´ comments.

Reviewer #3: (No Response)

**Results**

-Does the analysis presented match the analysis plan?

-Are the results clearly and completely presented?

-Are the figures (Tables, Images) of sufficient quality for clarity?

Reviewer #1: The results are clear and completely presented.

Reviewer #2: The revised version of the manuscript shows significant improvement and adequately addresses the reviewers´ comments.

Reviewer #3: I consider the authors' answers to all questions to be well documented, however, the addition of a figure showing sample locations and number of animals studies on a map of South Africa is still missing to illustrate Table 1.

For instance see Figure 1 in the article by Lane et al., 2014

https://pmc.ncbi.nlm.nih.gov/articles/PMC4159300/pdf/pone.0107038.pdf

**Conclusions**

-Are the conclusions supported by the data presented?

-Are the limitations of analysis clearly described?

-Do the authors discuss how these data can be helpful to advance our understanding of the topic under study?

-Is public health relevance addressed?

Reviewer #1: No comment

Reviewer #2: The revised version of the manuscript shows significant improvement and adequately addresses the reviewers´ comments.

Reviewer #3: (No Response)

**Editorial and Data Presentation Modifications?**

Reviewer #1: no comment

Reviewer #2: The revised version of the manuscript shows significant improvement and adequately addresses the reviewers´ comments.

Reviewer #3: (No Response)

**Summary and General Comments**

Reviewer #1: the manuscript has been improved so I have no more concern about it.

Reviewer #2: The revised version of the manuscript shows significant improvement and adequately addresses the reviewers´ comments.

Reviewer #3: (No Response)

PLOS authors have the option to publish the peer review history of their article (what does this mean? ). If published, this will include your full peer review and any attached files.

**Do you want your identity to be public for this peer review?** For information about this choice, including consent withdrawal, please see our Privacy Policy .

Reviewer #1: No

Reviewer #2: No

Reviewer #3: No

**Figure resubmission:**
---

## [Editor Report · Decision Letter 2]

5 Nov 2025

Response to Reviewers
Revised Manuscript with Track Changes
Manuscript

Shaden Kamhawi

co-Editor-in-Chief

Paul Brindley

co-Editor-in-Chief

**Additional Editor Comments:**
**Reviewers' comments:**
**Figure resubmission:**

**Reproducibility:** To enhance the reproducibility of your results, we recommend that authors of applicable studies deposit laboratory protocols in protocols.io, where a protocol can be assigned its own identifier (DOI) such that it can be cited independently in the future. Additionally, PLOS ONE offers an option to publish peer-reviewed clinical study protocols. Read more information on sharing protocols at https://plos.org/protocols?utm_medium=editorial-email&utm_source=authorletters&utm_campaign=protocols

---

## [Editor Report · Decision Letter 3]

16 Nov 2025

Dear Dr Cossu,

We are pleased to inform you that your manuscript 'Epidemiological baseline of Brucella spp. in South African wildlife' has been provisionally accepted for publication in PLOS Neglected Tropical Diseases.

Best regards,

Stuart D. Blacksell

Section Editor

Shaden Kamhawi

co-Editor-in-Chief

Paul Brindley

co-Editor-in-Chief

---

## [Editor Report · Acceptance letter]

Dear Dr Cossu,

We are delighted to inform you that your manuscript, "Epidemiological baseline of Brucella spp. in South African wildlife," has been formally accepted for publication in PLOS Neglected Tropical Diseases.

Best regards,

Shaden Kamhawi

co-Editor-in-Chief

Paul Brindley

co-Editor-in-Chief
